# Functional Expression and One-Step Protein Purification of Manganese Peroxidase 1 (rMnP1) from *Phanerochaete*
*chrysosporium* Using the *E. coli*-Expression System

**DOI:** 10.3390/ijms21020416

**Published:** 2020-01-09

**Authors:** Angel De La Cruz Pech-Canul, Javier Carrillo-Campos, María de Lourdes Ballinas-Casarrubias, Rosa Lidia Solis-Oviedo, Selena Karina Hernández-Rascón, León Raúl Hernández-Ochoa, Néstor Gutiérrez-Méndez, Antonio García-Triana

**Affiliations:** 1CONACyT—Faculty of Chemical Sciences, Autonomous University of Chihuahua, Campus II, Chihuahua 31125, Mexico; angelpechcanul@gmail.com; 2Faculty of Chemical Sciences, Autonomous University of Chihuahua, Campus II, Chihuahua 31125, Mexico; jc890917@hotmail.com (J.C.-C.); mballinas@uach.mx (M.d.L.B.-C.); a291849@uach.mx (S.K.H.-R.); lhernandez@uach.mx (L.R.H.-O.); ngutierrez@uach.mx (N.G.-M.); 3Healthy Life Importadora, Exportadora Y Productora De Alimentos Y Bebidas, Sociedad De Responsabilidad Limitada De Capital Variable, Chihuahua 31125, Mexico; solisoviedo@gmail.com

**Keywords:** manganese peroxidase, MnP1, recombinant, codon-optimization, *Phanerochaete*, *Escherichia coli*, inteins, chitin binding domain (CBD), protein purification, affinity chromatography

## Abstract

Manganese peroxidases (MnP) from the white-rot fungi *Phanerochaete chrysosporium* catalyse the oxidation of Mn^2+^ to Mn^3+^, a strong oxidizer able to oxidize a wide variety of organic compounds. Different approaches have been used to unravel the enzymatic properties and potential applications of MnP. However, these efforts have been hampered by the limited production of native MnP by fungi. Heterologous expression of MnP has been achieved in both eukaryotic and prokaryotic expression systems, although with limited production and many disadvantages in the process. Here we described a novel molecular approach for the expression and purification of manganese peroxidase isoform 1 (MnP1) from *P. chrysosporium* using an *E. coli*-expression system. The proposed strategy involved the codon optimization and chemical synthesis of the MnP1 gene for optimised expression in the *E. coli* T7 shuffle host. Recombinant MnP1 (rMnP1) was expressed as a fusion protein, which was recovered from solubilised inclusion bodies. rMnP1 was purified from the fusion protein using intein-based protein purification techniques and a one-step affinity chromatography. The designated strategy allowed production of an active enzyme able to oxidize guaiacol or Mn^2+^.

## 1. Introduction

Manganese peroxidases (MnP, E.C.1.11.1.13) were first described as a part of the lignin degrading system from the white-rot fungi *Phanerochaete chrysosporium* [1]. These enzymes have a heme prosthetic group, are H_2_O_2_ dependent, and catalyse the oxidation of Mn^2+^ to Mn^3+^ [2,3,4]. All known white-rot fungi produce MnP enzymes, enabling their capacity to degrade lignin. Indeed, it is due to this enzyme that fungi are the best known microorganism for degrading lignin polymer [5]. Lignin is an abundant biopolymer that plays a key role in supporting the growth of large and tall vascular plants. Its structure is a three-dimensional polymer network connected by several acid-resistant C-C linkages, consequently it is only partly degraded to monomeric compounds by hydrolysis but mostly degraded by oxidation [6]. The MnP catalytic cycle involves the cleavage of a molecule of H_2_O_2_ with the subsequent oxidation of the heme group in the enzyme structure. Then, Mn^2+^ is oxidized to Mn^3+^, a strong oxidizer which has to be stabilized by organic acids such as oxalate [7]. The Mn^3+^−organic acid complex formed during the reaction acts as a diffusible oxidant able to oxidize lignin and several xenobiotic compounds.

MnP has been broadly researched due to its enzymatic properties and its potential industrial applications. However, its extensive use is mainly hampered by two main intrinsic properties: the limited production of the enzyme by *P. chrysosporium* and its low stability [8]. Different approaches have been explored in order to optimize its enzymatic properties, for example site-directed mutagenesis was performed to obtain mutated versions of MnP that are more resistant to H_2_O_2_ [9], or less susceptible to elevated temperatures and/or pH [10]. On the other hand, several modifications of the growth conditions and culture media composition have been explored in order to improve the amount of enzymes produced by fungi. Such modifications included the immobilization of fungal cells [11], incubation at different ranges of temperature or pH, addition of Tween 80 or cofactors to the culture media, nitrogen limitation, growth on solid media instead of liquid cultures [12,13,14,15], and so on. Moreover, heterologous expression of MnP has been achieved in *Escherichia coli* [16,17,18], using the baculovirus expression system [19], in *Pichia pastoris* [20,21,22,23,24], in *Aspergillus* [8,25,26], and in a cell free system [27,28].

In the present work, a novel molecular approach was used to obtain a functional recombinant MnP1 (rMnP1) from *P. chrysosporium*. The MNP1 gene was codon-optimized and chemically synthesized for its heterologous expression in the *E. coli* expression system. Currently, this expression system represents a fast, efficient, and cheap strategy to express and recover proteins from either eukaryotic or prokaryotic sources. Recombinant protein was purified using an intein (INTervening protEINs) self-cleavage system. This self-splicing system negates the need for adding protease to catalyse release of the protein of interest [29]. For nearly two decades, these systems have shown their effectiveness as a quick method of purification with considerable yields of purified recombinant proteins [30,31,32]. To the best of the authors’ knowledge, the use of a synthetic, codon-optimized MnP1 gene combined with its heterologous expression in an *E. coli* host and intein-based protein purification techniques to achieve a purified rMnP1 has not been previously reported elsewhere.

## 2. Results

### 2.1. Construction of Recombinant Expression Vector pTXB1-MnP1

Synthetic *MnP1* gene was cloned into the pTXB1 vector using *Nde*I/*Sap*I restriction sites and delivered into *Escherichia coli* DH5α cells. Construct, termed pTXB1-MnP1, contains a fusion between the C-terminus of rMnP1 and the intein tag, which conveniently contains a chitin binding domain (CBD) for the affinity purification of the fusion protein on a chitin resin column. The overall process for the construction of pTXB1-MnP1 is summarized in the diagram depicted in Figure 1. Transformed clones were confirmed by colony PCR (Figure 2) with an expected band of 1100 bp. The correct nucleotide sequence of synthetic MnP1 gene was further confirmed by DNA sequencing. The coding sequence of the synthetic MnP1 gene was codon-optimized for better expression in *E. coli*. Its sequence encodes for a protein of 356 amino acids which is identical to isoform 1 of MnP from *Phanerochaete chrysosporium*.

### 2.2. Expression of Fusion Protein rMnP1−Intein−CBD

The construct pTXB1-MnP1 was used to transform *E. coli* T7 shuffle competent cells. This engineered strain constitutively expresses the disulfide bond isomerase (DsbC) which promotes disulfide bond formation in the cytoplasm. Additionally, DsbC promotes the correction of miss-oxidized proteins into their correct form by serving as a folding chaperone for proteins that do not require disulfide bonds. The expression of fusion protein rMnP1-intein-CBD and its solubilisation were tested with three different IPTG concentrations (0.1, 0.4, and 0.8 mM) at 37 °C for 12 h. After induction, cells were collected by centrifugation and suspended in lysis buffer. Sonication alternated with cooling in ice water was applied to the samples in order to enhance solubilisation. The cell-lysate was centrifuged at 15,000× *g* for 15 min at 4 °C. Soluble and insoluble fractions were both analysed by SDS-PAGE (Figure 3). Fusion protein (~66.5 kDa) was negligible in the soluble fractions of the three different conditions tested for induction. On the other hand, all insoluble fractions had an intense signal at the expected size of the fusion protein. This suggested that the protein fusion was stored in inclusion bodies, in a similar way as previously reported for other manganese peroxidases [16,17].

The protein expression was attempted to optimize with a fixed concentration of 0.1 mM IPTG. Assays to improve expression levels were conducted by lowering the incubation temperature or by varying induction time. However, the expression of the fusion protein was negatively affected by tested conditions [33]. Therefore, conditions for expression of fusion protein were fixed to induction with 0.1 mM IPTG at 37 °C for 12 h.

### 2.3. Solubilisation and Purification of rMnP1

For the solubilisation of proteins from inclusion bodies, different concentrations of urea (0.5, 1, 2, 4, or 8 M) were tested in the denaturing buffer (20 mM Tris-HCl pH 8.5; 500 mM NaCl; 1 mM EDTA). The solubilisate was incubated in ice for six h and further analysed by SDS-PAGE. The presence of the fusion protein in soluble form was confirmed, however, no obvious spot differences were observed between 4 or 8 M urea treatments (Figure 4). Concentrations below 4 M urea were also tested; however, these conditions failed to solubilisate inclusion bodies [33]. Thus, solubilisation for protein purification was conducted with 4 M, in order to diminish urea-induced changes that may affect protein rMnP1 structure.

Solubilisate extract containing fusion protein was applied to a chitin bead column at a flow rate of 0.5 mL min^−1^. The fusion protein rMnP1-intein-CBD was efficiently bound to the chitin column due to the chitin binding domain (Figure 5, lane **4**). The column was washed with 45 column volumes (cvs) of column buffer to remove unretained proteins (Figure 5, lane **5**), and self-cleavage of the fusion protein was induced with 3 cvs of cleavage buffer. After premature cleavage of the fusion protein was discarded (Figure 5, lane **6**), the column was sealed and incubated at room temperature for 36 h in order to promote self-cleavage of the fusion protein and the subsequent refolding of rMnP1. Ten cvs of column buffer were used to elute rMnP1 from the column, and each fraction (equivalent to one cv) was separately collected. The highest signal for rMnP1 was seen in the first 1–3 fractions (Figure 5, lane **7** and **8**). Furthermore, rMnP1 was negligible after the elution process (Figure 5, lane **9**).

### 2.4. Quantification and Protein Sequencing of rMnP1

Prior to quantification, DTT was removed from the sample during protein concentration in a Pierce Protein Concentrator PES, 30K MWCO (Thermo Scientific). The strategy followed in the present work allowed the production of ~25 mg of rMnP1 per each litre of IPTG-induced culture. In order to confirm the correct sequence of rMnP1, protein sequencing was accomplished through digestion of rMnP with trypsin and subsequent analysis of digested fragments by MALDI-TOF/TOF. The results confirm the designed amino-acid sequence of rMnP1, which is the same amino-acid sequence as the isoform 1 (MnP1) of *P. chrysosporium* (Accession number: Q02567. UniProtKB/Swiss database).

### 2.5. Enzyme Activity

In order to confirm the rMnP1 activity, standard assays previously reported for manganese peroxidases were carried out. The Varioskan Flash plate reader was used for measuring optical density. The machine is equipped with an automated dispensing unit, which has the advantage of accurately measuring the start point of each assay. The assays showed no changes in absorbance at the recommended final enzyme concentration [1,34]. It was found that enzyme activities for the oxidation of guaiacol were consistent with 2.8 or 1.4 µg mL^−1^ of rMnP1 (Figure 6A). The enzyme activity for these concentrations were 0.555 and 0.255 U L^−1^; with a similar specific activity of 0.196 and 0.179 U mg^−1^, respectively. Complementarily, assays for the oxidation of Mn^2+^ were also carried out in order to verify the capability of rMnP1 to oxidize Mn^+2^. These assays also confirmed the activity of rMnP1 (Enzyme activity 166.3 U L^−1^, specific activity 58.8 U mg^−1^) (Figure 6B).

## 3. Discussion

Manganese peroxidases have been extensively studied for more than a decade with many works focused on elucidating their enzymatic mechanisms [1,2,4,34]. Additionally, studies also reported the wide diversity of potential industrial applications they have [8,15,21]. The potential versatility of manganese peroxidases is probably the result of (1) a very high redox potential when compared to other peroxidases, and (2) the produced Mn^3+^-oxidant being able to oxidize several types of organic molecules [4]. Many efforts have been made towards improving the production of native MnP by fungi but have been met with limited success, hindering their widespread industrial use [11,12,13]. However, strategies for the overexpression and purification of MnP were accomplished in eukaryotic expression systems [19,20,21,22] and in the prokaryotic expression system of *Escherichia coli* [16,17,35]. In these studies, the authors overcame many hurdles, such as the potential contamination with other MnP and LiP isozymes or the formation of inclusion bodies that require further treatments to obtain an active enzyme.

In the present work we accomplished the expression of a functional rMnP1 by using a novel approach. In order to increase the quantity of expressing protein we first designed a synthetic MNP1 gene, which was codon-optimized for its best expression in *Escherichia coli*. Then, we created a fusion protein based on the intein-mediated purification system. The pTXB1 vector promotes the expression of a fusion protein rMnP1-intein-CBD, where the chitin binding domain (CBD) allowed the efficient purification of rMnP1 by a single-step affinity chromatography. Conveniently, the expression host, *E. coli* T7 shuffle, constitutively expresses the disulfide bond isomerase (DsbC), which has been proven to positively impact the production of recombinant MnP [28]. Additionally, the cleavage buffer contains dithiothreitol (DTT), which promotes the self-excision of the target protein. DDT is regularly used to reduce or to prevent the intermolecular interaction forming among cysteine residues of proteins. With this approach, we produced ~25 mg of purified rMnP1 from L^−1^ of induced culture. The amount of purified rMnP1 was remarkably higher than previous report for *E. coli* (275 µg L^−1^ in [17]) or even for some reports in eukaryotic expression systems (~550 µg L^−1^ in [23], 5 mg L^−1^ in [25]). Conversely, Conesa and coworkers [36] reported a higher production of recombinant MnP1 in *Aspergillius niger* by adding hemin or haemoglobin to the culture medium, yielding ~66 mg L^−1^ and ~100 mg L^−1^, respectively. Similar results were also reported in *Pichia pastoris* (~148 mg L^−1^) [24] and in *A. niger* (no yield reported) [26]. However, these studies involved the use of expensive heme additives for the production of a functional enzyme. Finally, the enzyme activity of the purified rMnP1 was confirmed by standard assays. The purified rMnP1 is able to oxidize guaiacol using a previously reported protocol for native MnP [1,2,3]. The assays were reproducible with an enzyme activity nearly proportional to the amount of rMnP1 added to the reaction (Figure 6A). Additionally, their specific activity values were comparable in agreement with a pure enzyme preparation. Complementarily, the capability of rMnP1 to oxidize Mn^2+^ was also verified. This assay is recommended for purified enzymes, which have less contaminant metals, such as Fe or Cu, present in crude preparations that inhibit activity. The progress of the reaction is consistently measured during the first 30-s (Figure 6B), as reported previously [37].

Other experimental approaches were attempted in order to optimize our strategy. For example, the expression of the fusion protein rMnP1-intein-CBD, under the control of T7 promoter, caused the formation of inclusion bodies, a drawback we tried to overcome by lowering expression levels with no remarkable results [33]. The incorporation of physical disruption methods, such as sonication and vortex, was insufficient to disaggregate inclusion bodies; consequently, the solubilisation of inclusion bodies was achieved with 4 M urea (Figure 5). Another point to consider is the proper refolding of MnP after recovering from inclusion bodies [16,17,28,35]. Currently there is no single method of refolding that fits for all proteins. The refolding of recombinant protein from bacterial inclusion bodies is still a challenge for the correct expression of proteins from any source. In this work, we did not include any additional steps for the in vitro refolding of rMnP1. It is likely that both the constitutive expression of DsbC by the *E. coli* host and the presence of DDT during the self-cleavage step promote the refolding of MnP1, since the purified enzyme was active. Hence, our strategy can be refined if combined with other improvements. For example, the production of a recombinant MnP in *E. coli* was favoured by the presence of calcium and hemin during in vitro refolding, generating an active enzyme with similar properties to those of the native MnP [17,28]. Furthermore, a recent work presented a strategy where the co-expression of a combination of chaperones, alongside the expression of DsbC, allowed production of a soluble form of recombinant MnP in *E. coli*. However, the recombinant MnP produced still had to be subject to in vitro maturation using hemin, ATP, and an ATP regeneration system in order to help chaperones to finish their folding cycle [35]. Another strategy that could be addressed is by simultaneously decreasing the temperature of expression (16 °C) and supplementing the culture medium with four chemical additives (Triton X-100, Tween-80, glycerol, and ethanol). This approach led to the production of a soluble, active form of MnP from *Irpex lacteus* using the *E. coli* expression system, which also prevented the formation of inclusion bodies [18]. Overall, the *E. coli* expression system has proven to be a promising alternative to the eukaryotic expression system for the expression of MnP, resulting in higher yields of active enzyme in fewer steps.

## 4. Materials and Methods

### 4.1. Synthetic MnP1 Gene

A search in the UniProtKB/Swiss database was performed to screen for MnP genes from *Phanerochaete chrysosporium*. At least four isoforms of the MnP gene were identified. Due to the complete information available for both gene and protein, the isoform 1 (MnP1, Accession number: Q02567) was chosen. The nucleotide sequence of cDNA encoding MnP1 was used to design a synthetic gene. The synthetic MnP1 gene was codon-optimized and chemically synthesized by DNA 2.0 (Newark, CA, USA). Codon optimization was performed with the GeneGPS™ program (DNA 2.0) using algorithms for high expression in *Escherichia coli*. The signal peptide (21-amino acids leader sequence) was excluded from the nucleotide sequence. For cloning into the expression vector, recognition sites for endonucleases *NdeI* and *SapI* were incorporated in the 5′ and 3′ ends of the synthetic sequence, respectively.

### 4.2. Strains, Vector, Chemicals, Media, Culture Conditions

*E. coli* DH5α strain (Invitrogen, Carlsbad, CA, USA) was used as the host strain for cloning and maintaining plasmids, whereas *E. coli* T7 shuffle strain (New England Biolabs, NEB) was used as the host for protein expression. Strains were cultured in Luria broth (LB) from Sigma. Liquid cultures were routinely incubated at 37 °C and 150 rpm. Culture media were supplemented with ampicillin (100 µg ml^−1^). Enzymes, expression vector pTXB1, and chitin resin were all included in the commercial IMPACT Kit (NEB). Primers were synthesized by Integrated DNA Technologies (IDT). Other reagents used in experimental procedures were purchased from Sigma.

### 4.3. PCR Conditions, Cloning, Transformation, and DNA Sequencing

The synthetic MNP1 gene was delivered cloned in pJ401:119209 (DNA 2.0). The synthetic gene was amplified from pJ401:119209 by PCR using Amplitaq Gold 360 Master Mix (Applied Biosystems) with specific primers F1: 5′- GGTGGTCATATGGCAGTTTGTCCGGATG-3′ and R1: 5′- GGTGGTTGCTCTTCCGCACGCCGGACCATTGAAT-3′. The incorporated *NdeI* and *SapI* restriction sites, respectively, are underlined. Amplification was performed on a Bio-Rad thermal cycler (Bio-Rad, model: T100) with 10 min at 94 °C for the initial denaturation step followed by 35 cycles of 1 min at 94 °C, 1 min at 68 °C, 1.5 min at 72 °C for extension with a final extension of 7 min at 72 °C. Purified PCR product and pTXB1 vector (IMPACT Kit, NEB) were both digested with *NdeI* and *SapI*. Digested fragments were purified with Illustra GFX (General Electric) and ligated with the T4 DNA ligase (PROMEGA). The ligation mix was used to transform competent cells of *E. coli* T7 shuffle. The construct pTXB1-MnP1, harbouring synthetic MNP1 gene was screened by colony PCR and its sequence was verified by commercial DNA sequencing (MACROGEN).

### 4.4. Expression of Fusion Protein rMnP1−Intein−CBD in E. coli

For expression of the fusion protein consisting of recombinant MnP1 (rMnP1), intein, and a chitin binding domain (rMnP1−intein−CBD), five mL of LB, supplemented with 100 µg mL^−1^ of ampicillin (LB_amp_), was inoculated with single colonies of *E. coli* T7 shuffle pTXB1-MnP1 and incubated overnight. Then, 25 mL of LB_amp_ was inoculated with 2 mL of the overnight culture and incubated until the OD_600nm_ reached 0.5. One litre of LB_amp_ was inoculated with the whole 25-mL culture and incubated until OD_600nm_ again reached 0.5. Next, protein expression was induced with the addition of 0.1 mM isopropyl-β-D thiogalactopyranoside (IPTG) and incubated for 12 h. Induction was tested with 0.4 or 0.8 mM IPTG added after 4, 8, or 24 h of incubation time. Cells were harvested by centrifugation at 3000× *g* at 4 °C for 30 min. About 6 g of wet cell pellet was recovered from 1 L of culture. From there, 1.7 g of pellet was re-suspended in 15 mL of lysis buffer (20 mM Tris-HCl pH 8.5; 500 mM NaCl; 1 mM EDTA; 0.1% *v*/*v* Triton X-100; 20 µM PMSF) and sonicated on ice. The lysate was centrifuged at 15,000× *g* for 30 min at 4 °C. Both soluble and insoluble fractions were analysed by SDS-PAGE.

### 4.5. Solubilisation of Fusion Protein rMnP1-Intein-CBD from Inclusion Bodies

The insoluble fraction (about 1 g) was resuspended in 5 mL of cold denaturation buffer (20 mM Tris-HCl pH 8.5; 500 mM NaCl; 1 mM EDTA) containing different concentrations of urea (0.5, 1, 2, 4, or 8 M) and incubated on ice for 5 h. The mixture was centrifuged at 12,000× *g* for 15 min at 4 °C. The resulting pellet was re-suspended in deionized water. Supernatant and resuspended pellet from the treated insoluble fraction were both analysed by SDS-PAGE in order to confirm the presence of a soluble form of the fusion protein.

### 4.6. Purification of rMnP1 by One-Step Affinity Chromatography and Intein Self-Cleavage

A column volume of 4 mL of chitin beads was equilibrated with column buffer (20 mM Tris-HCl pH 8.5; 500 mM NaCl; 1 mM EDTA; 4 M Urea). Supernatant containing the soluble form of the fusion protein was passed through the column at a flow rate of 0.5 mL min^−1^. Unbounded proteins were washed with 45 column volumes (cvs) of column buffer, followed by 3 cvs of cleavage buffer (20 mM Tris-HCl pH 8.5, 500 mM NaCl, 1 mM EDTA, 50 mM dithiothreitol (DTT)). Then, the column was sealed and incubated for 36 h at room temperature. rMnP1 was eluted from the column with 10 cvs of column buffer without urea, fractions of one column volume were collected separately.

### 4.7. Quantification and Protein Sequencing of rMnP1

Protein concentration was determined by the Bradford assay using serum albumin as standard. Prior to quantification, DTT was removed from the eluted fractions while rMnP1 was concentrated in a Pierce Protein Concentrator PES, 30K MWCO (Thermo Scientific). The correct amino acid sequence of rMnP1 was verified by mass spectrometry using MALDI-TOF/TOF carried in an AutoFlexSpeedTM TOF/TOF Burker spectrophotometer.

### 4.8. Enzyme Assays

Oxidation of guaiacol and oxidation of Mn^2+^ assays were both performed and measured by spectrophotometry as described previously [1,2,3,37]. Briefly, the oxidation of guaiacol was measured as the change in absorbance followed by the tetraguaiacol formation. The reaction mixture in a total volume of 200 µL contained rMnP1 enzyme, sodium succinate (500 mM, pH 4.5), guaiacol (100 mM), and MnSO_4_ (50 mM). Reactions were initiated by the addition of H_2_O_2_ (10 mM). The increase in absorbance at 465 nm was recorded at 5-s intervals for 10 min.

For the oxidation of Mn^2+^ assays, the oxidation of Mn^2+^ to Mn^3+^ was determined by measuring the absorbance change at 270 nm [38]. The reaction mixture, in a total volume of 200 µL, was comprised of rMnP1 enzyme, sodium malonate (50 mM, pH 4.5), and MnSO_4_ (100 mM). The Mn^3+^ produced forms a transiently stable complex with malonic acid with a characteristic absorbance at 270 nm. Reactions were initiated by the addition of H_2_O_2_ (100 mM). The increase in absorbance was monitored at 5-s intervals during the first 30 s of reaction. The enzyme activity was defined as follows: one unit (U) of enzyme oxidizes one µmol of guaiacol or Mn^2+^ per min. Optical densities (ODs) were measured spectrophotometrically at 25 °C with Varioskan Flash plate reader (Thermo Scientific) equipped with an automated dispensing unit. The enzyme was omitted in the reference assays. Assays were carried out in triplicate.

## 5. Conclusions

The molecular approach carried out in this work for the heterologous expression of a synthetic MnP1 gene in a prokaryote host, allowed for the production of a functional rMnP1 enzyme. This report makes available a novel strategy that hinges on a single-step purification process; such a strategy can be further scaled up for its exploitation in the research field or for biotech purposes. This approach seems promising for the expression and purification of MnP; however, it still needs additional improvements in order to increase its efficiency. Furthermore, structural and kinetic analysis of rMnP1 also needs to be investigated.

## Figures and Tables

**Figure 1 ijms-21-00416-f001:**
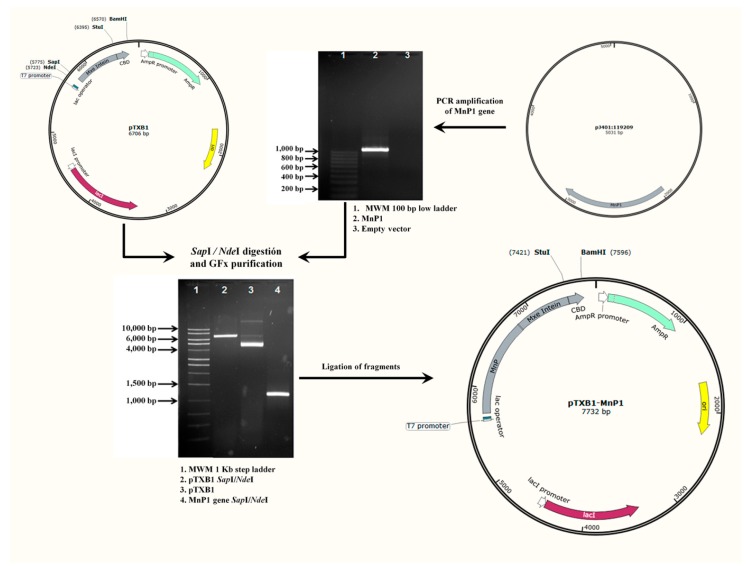
Schematic representation for the construction of the recombinant plasmid pTXB1-MnP1.

**Figure 2 ijms-21-00416-f002:**
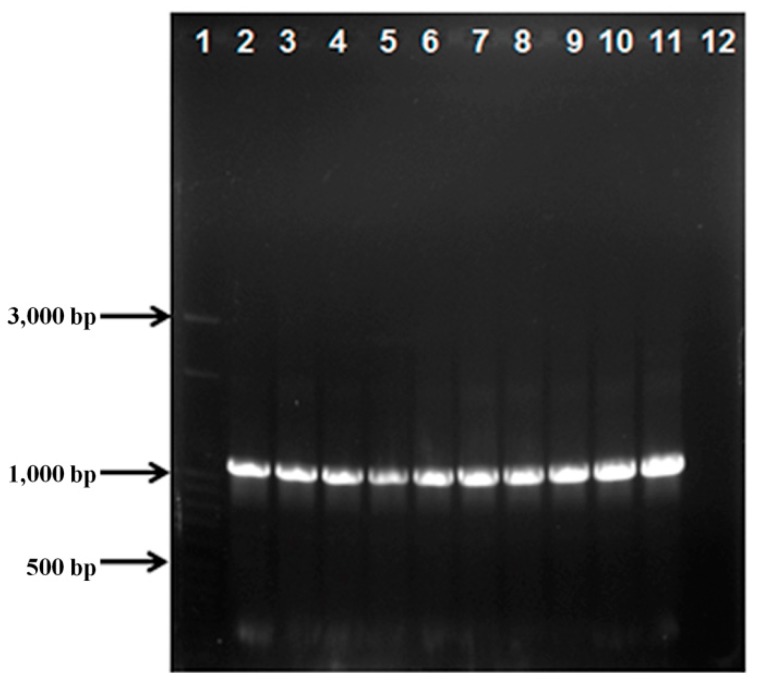
Colony PCR for pTXB1-MnP1 candidates. 1. MWM 50 bp step ladder; 2−11. Candidates; 12. Empty vector.

**Figure 3 ijms-21-00416-f003:**
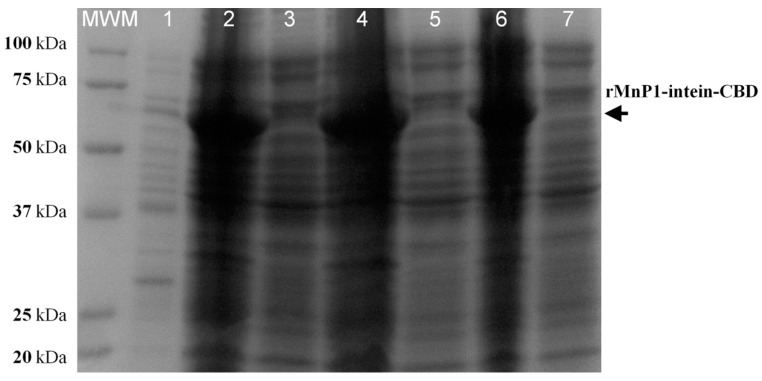
SDS-PAGE with uninduced and isopropyl-β-D thiogalactopyranoside (IPTG)-induced cells of *E. coli* T7 shuffle expressing the protein fusion rMnP1−Intein−CBD. When cultures reached an OD_600nm_ of 0.5, they were induced with different concentrations of IPTG and incubated at 37 °C for 12 h: induction with 0.1 mM IPTG, insoluble (lane **2**) and soluble (lane **3**) fractions; induction with 0.4 m M IPTG, insoluble (lane **4**) and soluble (lane **5**) fractions; induction with 0.8 mM IPTG, insoluble (lane **6**) and soluble (lane **7**) fractions. Lane **1**: Uninduced with IPTG. MWM: Kaleidoscope Protein Standard.

**Figure 4 ijms-21-00416-f004:**
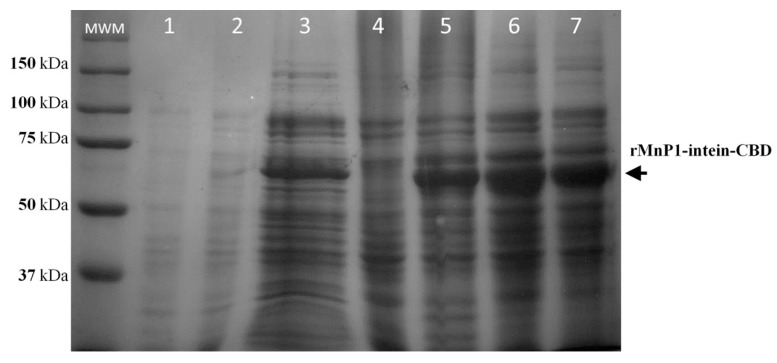
Solubilisation of MnP1-Intein-CBD fusion from inclusion bodies. SDS–PAGE analysis of the insoluble fraction treated with buffer containing 4 or 8 M urea. When cultures reached an OD_600nm_ of 0.5, they were induced with 0.1 mM of IPTG and incubated at 37 °C for 12 h. Lane **1**: Uninduced cells. Lane **2**: Induced cells. Lane **3**: Whole cell lysate. Lanes **4** and **5** correspond to soluble fraction and inclusion bodies, respectively. Inclusion bodies were treated with 4 M or 8 M urea (Lane 6 and 7, respectively). MWM: Kaleidoscope Protein Standard.

**Figure 5 ijms-21-00416-f005:**
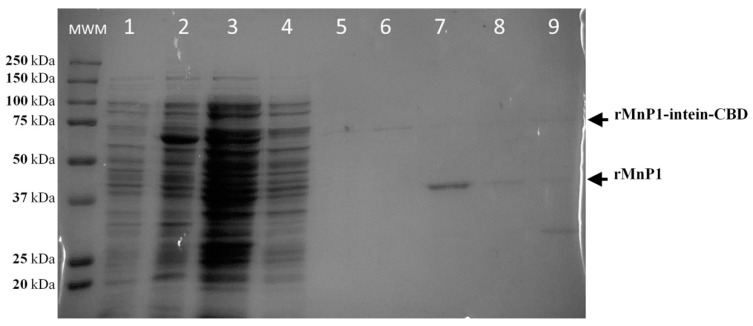
Purification of rMnP1. SDS–PAGE analysis for the purification of rMnP1 by one-step affinity chromatography. When cultures reached an OD_600nm_ of 0.5, they were induced with 0.1 mM of IPTG and incubated at 37 °C for 12 h. Inclusion bodies, containing fusion protein rMnP1-intein-CBD, were solubilised with 4 M urea. Lane **1**: Uninduced cells. Lane **2**: Induced cells. Lane **3**: Inclusion bodies treated with 4M urea. Lane **4**: Chitin column flow-through containing unretained proteins. Flow-through after washing with column buffer (Lane **5**) and after treating with cleavage buffer (Lane **6**). Lanes **7**–**8**: Fractions containing rMnP1. Lane **9**: Flow-through after the elution process. MWM: Kaleidoscope Protein Standard.

**Figure 6 ijms-21-00416-f006:**
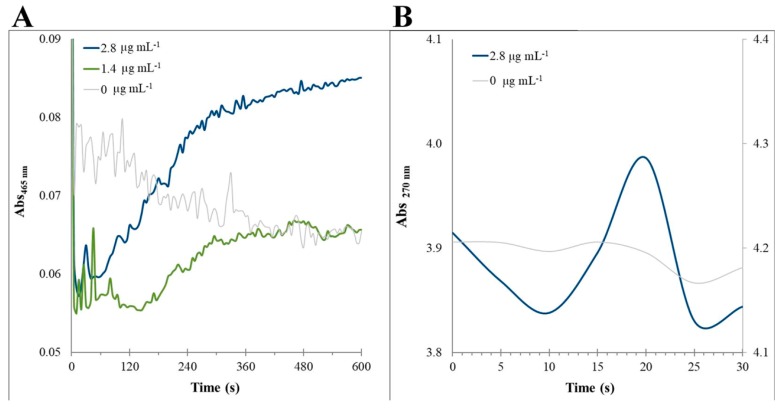
Standard assays for manganese peroxidases. Oxidation of guaiacol and oxidation of Mn^+2^ assays were both measured by spectrophotometry. In (**A**), the oxidation of guaiacol was measured as the change in absorbance at 465 nm. Reaction mixture contains rMnP1 (2.8 or 1.4 µg mL^−1^), sodium succinate (500 mM, pH 4.5), guaiacol (100 mM), and MnSO_4_ (50 mM). Reactions were initiated by addition of H_2_O_2_ (10 mM). For the oxidation of Mn^2+^ assays (**B**) the oxidation of Mn^2+^ to Mn^3+^ was followed by measuring the absorbance change at 270 nm. The reaction mixture comprises of rMnP1 enzyme (2.8 µg mL^−1^), sodium malonate (50 mM, pH 4.5), and MnSO_4_ (100 mM). Reactions were initiated by addition of H_2_O_2_ (100 mM). The final volume of each reaction mix was 200 µL. The enzyme was omitted (0 µg mL^−1^) in the reference assays. Plots are representative from each of the triplicate sets.

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
