# Peer review of "Functional Expression and One-Step Protein Purification of Manganese Peroxidase 1 (rMnP1) from Phanerochaete chrysosporium Using the E. coli-Expression System"

_ijms, 2020, doi:10.3390/ijms21020416_

Round 1

Reviewer 1 Report

This paper reports the expression of Manganese Peroxidase 1 (rMnP1) from Phanerochaete chrysosporium in a E. coli expression system as a fusion protein, which was recovered from solubilised inclusion bodies and then purified using intein-based affinity chromatography.

In my opinion science is good and results are convincing. The discussion is well-written and provides interesting and detailed information about expression and purification experiments. In addition, the yield of purified rMnP1 was higher than the previous reports for eukaryotic and also for other E.Coli expression systems. However, to improve this paper  it will be interesting to test the purified enzyme in a substrate such as lignin, for example in pulp bleaching to study it´s industrial applicability.

Author Response

Reviewer 1.

This paper reports the expression of Manganese Peroxidase 1 (rMnP1) from Phanerochaete chrysosporium in a E. coli expression system as a fusion protein, which was recovered from solubilised inclusion bodies and then purified using intein-based affinity chromatography.

In my opinion science is good and results are convincing. The discussion is well-written and provides interesting and detailed information about expression and purification experiments. In addition, the yield of purified rMnP1 was higher than the previous reports for eukaryotic and also for other E. coli expression systems. However, to improve this paper it will be interesting to test the purified enzyme in a substrate such as lignin, for example in pulp bleaching to study it´s industrial applicability.

A:

This communication aims to validate a novel strategy, involving heterologous expression and one-step purification of MnP1, in order to produce a functional recombinant enzyme. We consider that our results sustain the main objective of the study. Mainly, the assay of oxidation of Mn2+demonstrates the production of the Mn3+−malonate complex, which is a strong oxidant able to oxidize several organic compounds, including lignin.

Regarding all the possible industrial applications of MnP1 (References 5, 8, 15, 20); we consider the reviewer’s comment is very adequate. Lignin is a polyphenolic material formed by poly propanoid units, linked to each other in irregular order via ether linkages and C-C bonds. The effects of oxidative enzymes have been monitored commonly by measuring the consumption of a co-substrate such as oxygen, or studying the formation of radicals. The complete study merits a whole paper for its clear and deep description.

Reviewer 2 Report

In this manuscript, the authors described their work done in optimizing the production of Manganese peroxidase from P. chrysosporium. They used an E coli expression system followed by intein-based protein purification. Manganese peroxidase has significant applications in degrading lignin, an important component of wood.

There are a few concerns about the methods and results presented in the manuscript.

The authors mentioned that the nucleic acid sequence was codon optimized for expression in E coli. However, there is no mentioning of how the codon optimization was performed, ie the software used, or the methodology (eg. codon optimization parameters). The authors should add a short description on how it was performed.

In the results section, the authors described the verification of the enzyme expressed and purified. However, there is no mentioning of the yield obtained or comparison with other efforts to express the enzyme. Only the results of the enzyme was tested at standard concentrations. In the discussion section, it was mentioned that 25mg/L-1 of MnP1 was obtained, which was compared to published yield from relatively old articles. Are there more recent relevant works that represent the more state-of-art with regards to MnP1 expression? It appears that there are published reports achieving higher yields (100 mg/L) with A. niger (albeit with heme supplementation). The authors should further compare and discuss their results with respect to other published works to highlight the advantages of their system, as well as how it may be further improved in the future.

Author Response

Reviewer 2.

In this manuscript, the authors described their work done in optimizing the production of Manganese peroxidase from P. chrysosporium. They used an E coli expression system followed by intein-based protein purification. Manganese peroxidase has significant applications in degrading lignin, an important component of wood.

There are a few concerns about the methods and results presented in the manuscript.

… The authors mentioned that the nucleic acid sequence was codon optimized for expression in E coli. However, there is no mentioning of how the codon optimization was performed, ie the software used, or the methodology (eg. codon optimization parameters). The authors should add a short description on how it was performed….

A:

The codon optimization was carried out by the DNA2.O Company with their own GeneGPS™ program using algorithms for high expression in Escherichia coli.  We added that description to the ‘material and methods’ section.

… In the results section, the authors described the verification of the enzyme expressed and purified. However, there is no mentioning of the yield obtained or comparison with other efforts to express the enzyme. Only the results of the enzyme were tested at standard concentrations…

A:

This communication focuses on validating the strategy for the production of a functional rMnP1. We did not contemplate the biotechnological application of crude extracts. Both protein quantification and enzymatic activity were not verified in the different sub-stages of the process. We considered that a measurable enzymatic activity would be unlikely at those stages since rMnP1 is expressed as a chimeric protein (comprising MnP-Intein-CBD) which led to the formation of inclusion bodies. Therefore, we focused on strategies to improve the solubilisation of inclusion bodies and the subsequent purification process.

… In the discussion section, it was mentioned that 25mg/L-1 of MnP1 was obtained, which was compared to published yield from relatively old articles. Are there more recent relevant works that represent the more state-of-art with regards to MnP1 expression?...

A:

There are a few relatively recent studies regarding the expression and purification of MnP1 from P. chrysosporium, for example, we included a pair of recent studies where MnP1 was co-expressed with chaperones in the E. coliexpression system (Alfi, 2019) and in an E. colicell-free system (Ninomiya, 2014) but a yield was not reported in both studies. Nevertheless, in the discussion section we mentioned that our strategy (also based in an E. colisystem) could be further improved by combining some of these reported strategies.

Additionally, we added new references regarding the expression of MnP1 in eukaryotic expression system, and the expression/purification of a MnP from Irpex lacteususing the E. coliexpression system (Wang, 2016). See below.

 It appears that there are published reports achieving higher yields (100 mg/L) with A. niger(albeit with heme supplementation).

A:

We rephrased the cited reference of Conesa (2000), and added new, similar references of studies in A. niger(Cortés-Espinosa, 2011) and in P. pastoris(Jiang 2008).

The authors should further compare and discuss their results with respect to other published works to highlight the advantages of their system, as well as how it may be further improved in the future.

A:

In the discussion, we aim to highlight that our strategy allows the production of functional MnP1 using only a one-step protein purification system. Our strategy did not require: an extra step for in vitro refolding of MnP1 or the supplementation of heme additives to the culture medium, as reported previously.

We suggested that our strategy could be further improved by combining some of the reported strategies (Alfi, 2019; Ninomiya, 2014; Whitwam, 1996). In this regard, we added a new reference about the novel strategy employed for the expression and purification of a MnP from Irpex lacteususing theE. coliexpression system (Wang, 2016).